# A Novel Predictive Machine Learning Model Integrating Cytokines in Cervical-Vaginal Mucus Increases the Prediction Rate for Preterm Birth

**DOI:** 10.3390/ijms241813851

**Published:** 2023-09-08

**Authors:** Hector Borboa-Olivares, Maria Jose Rodríguez-Sibaja, Aurora Espejel-Nuñez, Arturo Flores-Pliego, Jonatan Mendoza-Ortega, Ignacio Camacho-Arroyo, Ramón Gonzalez-Camarena, Juan Carlos Echeverria-Arjonilla, Guadalupe Estrada-Gutierrez

**Affiliations:** 1Community Interventions Research Branch, Instituto Nacional de Perinatología Isidro Espinosa de los Reyes, Mexico City 11000, Mexico; 2PhD Program in Biological and Health Sciences, Universidad Autónoma Metropolitana, Mexico City 09310, Mexico; 3Department of Maternal-Fetal Medicine, Instituto Nacional de Perinatología Isidro Espinosa de los Reyes, Mexico City 11000, Mexico; mariajose.rodriguezs@yahoo.com; 4Department of Immunobiochemistry, Instituto Nacional de Perinatología Isidro Espinosa de los Reyes, Mexico City 11000, Mexico; aurora.espejel@inper.gob.mx (A.E.-N.); arturo_fpliego@yahoo.com.mx (A.F.-P.); 5Department of Bioinformatics and Statistical Analysis, Instituto Nacional de Perinatología Isidro Espinosa de los Reyes, Mexico City 11000, Mexico; santos_jonatan@hotmail.com; 6Unidad de Investigación en Reproducción Humana, Instituto Nacional de Perinatología, Facultad de Química, Universidad Nacional Autónoma de Mexico, Mexico City 11000, Mexico; camachoarroyo@gmail.com; 7Department of Health Sciences, Universidad Autónoma Metropolitana, Unidad Iztapalapa, Mexico City 09310, Mexico; ramongoca@gmail.com; 8Department of Electrical Engineering, Universidad Autónoma Metropolitana, Unidad Iztapalapa, Mexico City 09310, Mexico; jcea@xanum.uam.mx; 9Research Division, Instituto Nacional de Perinatología Isidro Espinosa de los Reyes, Mexico City 11000, Mexico

**Keywords:** preterm delivery, screening, artificial intelligence, inflammatory response, interleukin-2, cervical length

## Abstract

Preterm birth (PB) is a leading cause of perinatal morbidity and mortality. PB prediction is performed by measuring cervical length, with a detection rate of around 70%. Although it is known that a cytokine-mediated inflammatory process is involved in the pathophysiology of PB, none screening method implemented in clinical practice includes cytokine levels as a predictor variable. Here, we quantified cytokines in cervical-vaginal mucus of pregnant women (18–23.6 weeks of gestation) with high or low risk for PB determined by cervical length, also collecting relevant obstetric information. IL-2, IL-6, IFN-γ, IL-4, and IL-10 were significantly higher in the high-risk group, while IL-1ra was lower. Two different models for PB prediction were created using the Random Forest machine-learning algorithm: a full model with 12 clinical variables and cytokine values and the adjusted model, including the most relevant variables-maternal age, IL-2, and cervical length- (detection rate 66 vs. 87%, false positive rate 12 vs. 3.33%, false negative rate 28 vs. 6.66%, and area under the curve 0.722 vs. 0.875, respectively). The adjusted model that incorporate cytokines showed a detection rate eight points higher than the gold standard calculator, which may allow us to identify the risk PB risk more accurately and implement strategies for preventive interventions.

## 1. Introduction

Preterm birth is a major public health issue concerning perinatal mortality, long-term morbidity, and economic burden, with a worldwide prevalence of 5–18%, 90% of which occurs in developing countries [1,2]. Spontaneous preterm birth (SPB) is defined as the delivery of the fetus before 37 weeks of gestation, calculated by the last menstrual period or reliable first-trimester ultrasound, without any medical intervention to induce this outcome [3]. Although some previous evidence suggests that preterm labor is a heterogeneous condition triggered by multiple factors [4,5], the inflammatory response elicited at the maternal-fetal interface is considered a hallmark of the pathology. The increase in the local cytokine release that orchestrates the inflammatory response is observed in cases with intrauterine infection and also during spontaneous preterm labor that is not associated with infection, thereby described as the intrauterine inflammatory response syndrome [6].

In vivo, cytokines are part of the molecular network mediates innate immune and inflammatory responses [7]. All cytokines expressed locally in the cervix participate in complex interactions with prostaglandins and nitric oxide, which regulate the production of extracellular matrix proteases and other factors associated with the cervical shortening, rupture of membranes, and uterine contractions all leading to labor either at preterm or term [6,8,9]. An increase in the pro-inflammatory cytokines, IL-6 and IL-8, in the cervical-vaginal fluid has been associated with cervical effacement [10,11], while an increased TNF-α is associated with cervical ripening [12]. On the other hand, IL-10 concentration has been identified as the main anti-inflammatory cytokine involved in PB pathogenesis [13]. So far, the results of studies analyzing the relationship between cytokines and PB are inconsistent, and none have identified a biomarker that can accurately predict preterm delivery [7]. These inconsistencies can be explained by the complexity of the local inflammatory processes and the heterogeneity of the pathways that trigger PB.

There is a pathophysiological association between preterm premature rupture of membranes (PPROM) and PB, both triggered when the intrauterine or maternal environment is hostile. Moreover, PPROM is among the leading causes of preterm birth [14], probably due to the onset of an intrauterine infectious/inflammatory process, which produces an imbalance in cytokine production, disrupting the tight junctions of the membranes, and rupturing the amniotic sac [15]. Some cytokines associated with PPROM and PB in placental tissue are IL-1β, IL-6, IL-8, and transforming growth factor (TGF-β) [16].

Among the screening techniques to detect women at a high risk for PB, the most used measurement in clinical practice is cervical length (CL), obtained by vaginal ultrasound in the second trimester of gestation, with a detection rate of 50 to 70%. Also, some clinical calculators incorporate CL, gestational age, and obstetric history to provide a patient-specific risk for developing PB. The calculator that is considered the gold standard in clinical practice is the Fetal Medicine Foundation (FMF) calculator, having a detection rate for preterm spontaneous birth of <28 weeks = 75%, 28–30 weeks = 57%, 31–33 weeks = 46%, and 34–36 weeks = 24%, considering a false positive rate of 10% [17].

In recent years, artificial intelligence has been used in different healthcare fields to predict, prevent, diagnose, and monitor different pathologies, even in obstetrics [18]. Models using machine learning seem more accurate than risk calculators as these are used to analyze massive data and, through algorithms, identify patterns for making predictions [19]; furthermore, it has also been proposed that machine learning models can be helpful in personalized pregnancy management, especially in low- and middle-income countries [18]. The goal of identifying women at high risk for developing PB is to individualize the clinical follow-up and offer medical preventive strategies (e.g., progesterone) that reduce by up to 90% of the risk of preterm birth in women with a history of this outcome and by 42% in pregnant women with short cervix detected in second-trimester screening [20].

Although the role of cytokines as critical mediators in the inflammatory process that triggering labor has been demonstrated [21], very few predictive models consider their measurement in the PB screening process [22,23,24]. However, none have been implemented in clinical practice. Therefore, this study aimed to characterize the cytokine profile in cervical-vaginal mucus in pregnant women with low and high risk for PB and then to integrate them in a PB screening model using a predictive machine learning analysis.

## 2. Results

A total of 60 pregnant women were recruited, including 40 participants who were considered at low risk and 20 at high risk for PB. The low-risk group had a prevalence of spontaneous preterm birth before 37 gestation weeks of 7.5% versus 45% in the high-risk group. The maternal characteristics of both groups are shown in Table 1. The study groups were similar in age, maternal BMI, socioeconomic level, smoking, and gestational age at which CL was measured. As expected, CL was significantly lower in the group of women at high risk for PB (p = 0.02).

### 2.1. Cytokine Profile in Low and High Risk for PB

The mean concentration of the pro-inflammatory cytokines, IL-2, IL-6, and IFN-γ, was significantly higher in the high-risk group (*p* = 0.001). As regards of cytokines with anti-inflammatory function, an increase in the concentration of IL-4 and IL-10 was found in the high risk group for PB (*p* = 0.001). In contrast, the IL-1ra concentration was significantly lower (*p* < 0.01) compared to women in the low-risk group for PB group. Table 2, Figure 1.

### 2.2. Machine Learning Predictive Model

Two predictive models were generated by machine learning analysis. The full model, which included all predictor variables, showed a detection rate of 66%, with FPR = 12%, FNR = 28%, and an AUC of 0.722. The adjusted model, which only included variables with clearer statistically significant differences (Maternal age, CL, and IL-2 concentration) showed a detection rate of 87%, with FPR = 3.33%, FNR = 6.66%, and an AUC of 0.875; to compare our model with the performance of the calculator used by the FMF, we entered our data in the FMF’s online calculator (https://fetalmedicine.org/research/assess/preterm/cervix, accessed on 1 May 2023), obtaining the following results: a detection rate of 79% with FPR = 6.60% and FNR = 11.66% (Table 3, Figure 2). The relevance of each predictor in the adjusted model was 26% for maternal age, 38% for CL, and 36% for the IL-2 concentration (Figure 2).

## 3. Discussion

In this study, we first focused on the charactering the inflammatory cytokine profiles in cervical-vaginal mucus of pregnant women with high and low risk for PB, as classified according to the CL measured at the second trimester of gestation. In agreement with previous reports, a higher concentration of IL-2, IL-6, and IFN-γ was found in the cervical-vaginal samples of the high-risk group, and also a higher concentration of IL-1ra in the low-risk group [25,26,27].

On the other hand, the concentrations of IL-4 and IL-10 were found increased as well in the high-risk group, in contrast to what was expected according to the inflammatory pathogenesis of PB [28]. These differences could be explained by the complexity of the inflammatory processes involved in PB and owing to the different methodologies and designs used in other studies [29]. The elevation of anti-inflammatory cytokines in the high-risk group could a manifestation of a be a physiological attempt to moderate the inflammatory process and maintaining homeostasis; this hypothesis was previously proposed by Wang et al., as they found that IL-37 (an anti-inflammatory cytokine) was elevated in the fetal membranes of women with preterm labor, which they considers a response to stop the inflammatory process caused by IL-6 [30].

IL-6 is a critical mediator in infection and inflammation and one of the most studied biomarkers associated with cervical shortening in preterm labor. In our study, IL-6 was found to increase in cervical-vaginal samples from the high-risk group, which coincides with the findings reported by other studies [31]. Goepfert et al. found that the cervical IL-6 concentrations, measured at the 24 weeks of gestational age (wGA), were elevated in women who had preterm delivery before 32 wGA compared to women whose pregnancies were carried to term. And this finding was even more evident in women with a history of preterm delivery [31,32,33]. In our study, 40% of the women in the high-risk group had a history of a previous delivery before 37 wGA [32,33,34].

In our study, a higher concentration of IFN-γ was found in the group of patients at high risk for preterm delivery. Accordingly, in a recent report by Sandoval-Colin et al., it was found a positive correlation between elevated IFN-γ, TNF-α, IL-1β or IL-6 and the onset of labor [22] in what appears to be a functional “maturation” of the immune system owing to the inflammatory response. Additionally, it has been described that the activation of Th1 cells may increase the secretion of TNF-α, INF-γ, and IL-1β in the fetal membranes and the amniotic fluid in preterm labor [35].

IL-1ra belongs to the IL-1 family of cytokines with an anti-inflammatory action [36]; our study found the concentration of IL-1ra to be higher in patients at low risk for PB. To our knowledge, there are not previous reports about changes of the cervical concentration of this cytokine in the pathogenesis of preterm delivery; however, our IL-1ra results could be well compared to those of the soluble IL-6 receptor (sIL-6r) that has been associated with a reduction in the risk of preterm delivery (RR 0.4 CI 95% 0.15–0.80) [37,38].

IL-2 is crucial for maintaining the immune homeostasis, and a correlation has been previously demonstrated between elevated IL-2 levels and chronic inflammation [39]. Few studies have demonstrated the relevance of IL-2 and its receptor in the pathogenesis of preterm labor; a possible explanation is that in normal pregnancy, the concentrations of IL-2 and its receptor are relatively low and difficult to measure. But in the case of preterm labor, an increase of this interleukin has been demonstrated indeed, which could imply the chronicity of an inflammatory process and not the acute response as in the case of IL-6 [40].

In most previous publications concerning cytokines in cervical-vaginal fluid, the elevated concentrations of IL-10 have been proposed as a protective factor for preterm delivery [41,42], because it belongs to the group of cytokines with anti-inflammatory activity. In our study, the highest concentrations were observed in the high-risk group, which is in agreement with the study by Vogel et al. who reported that the elevated IL-10 concentrations could also be associated with an increased risk of preterm delivery (RR 3.1 CI 95% 0.96–9.7) [43] becoming a response to an inflammatory mechanism that was previously initiated [41].

In the second phase of our study, we applied an artificial intelligence model for PB prediction using machine learning (ML) by incorporating quantification of cytokines because these are the crucial mediators of the inflammatory process observed in preterm labor. The main areas that may benefit from ML techniques in the medical field are diagnosis and outcome prediction; ML can transform how medicine works [19,44]. The use of AI methods in medical care could facilitate personalized pregnancy management and improve public health, especially in low- and middle-income countries. ML allows us to analyze interactions between variables different from what we are conventionally used to, overcoming limitations such as sample size and data distribution. In our study, two predictive models were performed: the full and the adjusted models. The full model included all predictor variables recorded (obstetric history, CL, and the concentration of all the cytokines measured). The ML analysis allowed us to choose only the variables with the highest predictive significance to build and train the adjusted model that only included maternal age, CL, and IL-2 concentration. Our results demonstrated that the adjusted model, even when it included a smaller number of variables, had a better predictive performance; this is possible because of the type of analysis used, in which some variables of the entire model are eliminated to improve performance [44,45,46].

In most studies involving cytokines in high- and low-risk groups for PB, the classification only considers the obstetric history [47]. Thus one of the main strengths of our work is that we studied the cervical-vaginal fluid concentration of the principal cytokines involved in the inflammatory process in patients classified as high and low risk for PB by CL (the gold standard in current screening and a variable which itself indicates the onset of cervical shortening and, therefore, the phase prior to the onset of labor). Additionally, we performed the screening at the appropriate weeks of gestation in accordance with the international guidelines [48] and it was not biased by interventions such as cerclage or progesterone before measurement. Furthermore, the presence of vaginal infection during sampling for cytokine determination was also ruled out.

Another contribution of our work is comparing the FMF calculator and the predictive model obtained by the Random Forest analysis. With this analysis, we demonstrate that if we add the measurement of IL-2 to the prediction model, we can increase, by 8 percentage points, the detection rate reported by the gold standard in clinical practice, which would help to identify more accurately those cases at high risk for PB, allowing us to implement preventive medical strategies (e.g., progesterone) that can offer efficiency up to 90% [49,50].

The main limitation of our study is the number of patients included; however, it was possible to identify significant statistical differences between groups. As regards the construction of the model, this fact is compensated by the nature of the analysis performed since the Random Forest analysis builds the ideal model and then replicates it thousands of times to test its efficiency [51]. However, we consider that a more significant number of patients is required to strengthen the model to be used as a reference for PB prediction in clinical practice [52].

As we aimed to design an effective screening model for PB with the inclusion of clinically applicable markers based on cost, availability, efficacy, and timeliness of the test, the next step is undoubtedly to perform a cost/benefit analysis to assess whether adding the IL-2 measurement to the screening model is endorsed by the resources saved with the risk detection and prevention of PB.

Our results contribute to a better understanding of the cervical-vaginal inflammatory network elicited in patients classified as high and low risk for preterm delivery in accordance with to the current screening gold standard. The prediction model that incorporates cytokines in cervical-vaginal mucus showed a DR 8 percentage points higher than the gold standard used in clinical practice, with a lower FNR. The increase in DR with the reduction in FNR may allow us to identify women at risk for PB more accurately and implement strategies to prevent this outcome. Our work could represent a significant advance screening to reducethe prevalence of PB, a goal that has not yet been achieved in the last 30 years.

## 4. Materials and Methods

### 4.1. Ethics Statement

The study was conducted at the Instituto Nacional de Perinatologia in Mexico City between January 2017 and December 2021. The protocol was approved by the Ethics, Research, and Biosafety Internal Review Boards (2017-2-69). Women who met the inclusion criteria were invited to participate, and they read and signed the informed consent form.

### 4.2. Study Population

Recruitment at convenience was performed by considering women who attended the Maternal-Fetal Medicine Department at the Instituto Nacional de Perinatologia, in Mexico City, to screen for PB with a singleton 18–23.6 weeks of gestation pregnancy. Clinical data that included age, weight, pregestational weight, socioeconomic status, smoking, history of preterm delivery, gestational age by date of last period as well corroborated by first trimester US, presence of urinary or vaginal infections, and use of antibiotics, and follow-up of pregnancy until its termination, were collected. Fetuses with structural alterations or women with a confirmed diagnosis of isthmic cervical incompetence were not included. The primary outcome was considered to be spontaneous preterm birth, defined as the delivery of the fetus before 37 weeks of gestation, calculated by the last menstrual period or reliable first-trimester ultrasound, without any medical intervention that produced this outcome [3].

Participants were classified as having high or low risk for developing PB according to the cervical length measurement, using a cut-off point of <20 mm in patients with a history of preterm delivery and <25 mm for patients without a history of PB [53]. Our institutional protocol was applied in high-risk patients by administering micronized progesterone, 200 mg, vaginally every 24 h, with a follow-up every 2–3 weeks. Progesterone treatment was initiated after being classified as a high-risk patient based on CL measurement. No patient was receiving progesterone before being classified.

### 4.3. Sample Collection

A cervical-vaginal mucus sample from the posterior vaginal fornix was obtained using a dacron polyester-tipped swab which was rinsed in collection buffer containing 1X PBS, 0.05% Tween-20, 1% BSA, and a protease Inhibitor Cocktail (Roche). Samples in the collection buffer were centrifuged at 3200 rpm, for 15 min at 4 °C, and the supernatants were stored at −80 °C until further cytokine analysis. Before preservation, an aliquot was tested to rule out subclinical vaginal or uterine infection by fresh examination and negative microbiological cultures.

### 4.4. Cervical-Vaginal Cytokine Quantification

Cytokine concentration in cervical-vaginal mucus was measured using the Luminex X-Map platform with the Bio-Plex Pro Human Cytokine 10-plex that includes IL-2, IL-4, IL-6, IL-8, IL-10, INF-γ, TNF-α, IL-1β, IL-12p70, and IL-1ra (Cat 12020756). The protocol was performed in accordance with he manufacturer’s instructions, and the value for each cytokine was expressed as pg/mL. The inter- and intra-assay variation was <10%.

### 4.5. Statistical Analysis

Data were analyzed with the SPSS software, version 24. Descriptive statistics were performed to characterize groups: for qualitative variables, frequency measures expressed in percentages were used, while for quantitative variables, measures of central tendency (mean, median), and measures of dispersion standard deviation (SD) were used. Data distribution was verified before the statistical analysis with a Kolmogorov–Smirnov test. Cytokine concentration was evaluated, and differences between groups were assessed using the Mann–Whitney U test. The Chi-square test was used to calculate the difference in proportions. No sample size calculation was performed beforehand, but the statistical power was calculated for all variables with significant differences to verify that it was greater than 80%.

### 4.6. Machine Learning Model

A Random Forest classifier first model was created to predict the incidence of preterm infants using 12 clinical variables (women’s age, weight, pregestational weight, socioeconomic status, smoking history, parity, previous preterm delivery, gestational age at screening (date of last period, corroborated by first-trimester US), precedent urinary or vaginal infections, use of antibiotics, and gestational age at the time of pregnancy resolution) and 10 cervical-vaginal cytokine determinations (IL-2, IL-4, IL-6, IL-8, IL-10, INF-γ, TNF-α, IL-1β, IL-12p70, and IL-1ra). Secondly to obtain an adjusted the model, we performed hyperparameter selection by out-of-bag error and simple cross-validation, resulting in 50 trees, 1 mtry (number of predictors considered in each split), and 1 as maximum depth.. The models were analyzed in a mathematical matrix to determine false positive rates (FPR) and false negative rates (FNR), their overall efficiency, as well as to compare the area under the ROC curves (AUC).

## Figures and Tables

**Figure 1 ijms-24-13851-f001:**
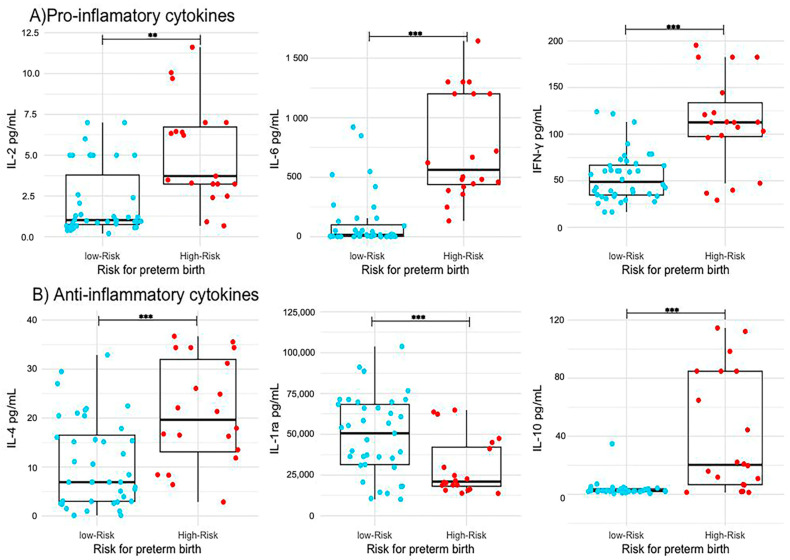
Differences in cytokines measured in cervical mucus. Boxplots showing (**A**) pro-inflammatory cytokines (IL-2, IL-6, and IFN-γ) and (**B**) anti-inflammatory cytokines (IL-4, IL1ra, and IL-10) with significant differences between women at low risk (*n* = 40) and high risk (*n* = 20) for preterm birth. ** *p* < 0.01, *** *p* < 0.001.

**Figure 2 ijms-24-13851-f002:**
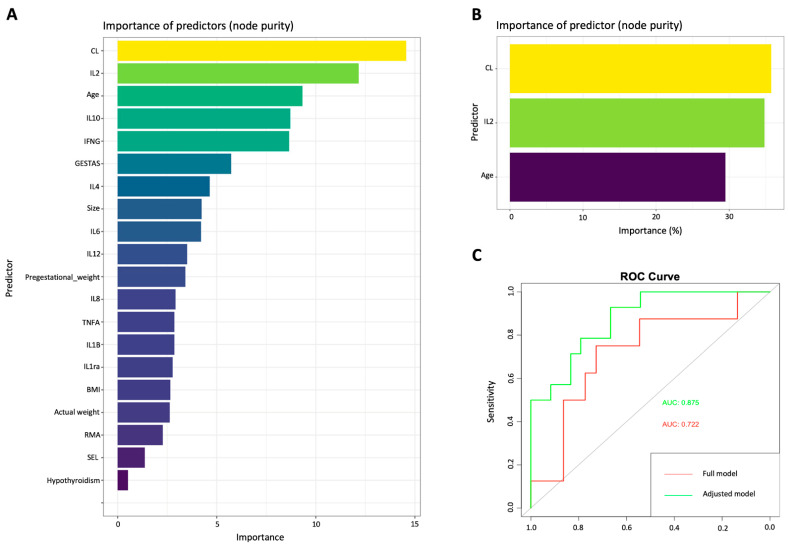
Performance of the models generated by machine learning. (**A**) Full model including all the variables studied as predictors. (**B**) Adjusted model including only the variables with the highest statistical relevance (CL, IL2, and maternal age). (**C**) ROC curves of the two proposed models: red, full model with an AUC = 0.722, and green, adjusted model with an AUC = 0.879. CL: Cervical Length, BMI: Body Mass Index, RMA: Risk Maternal Age, IL: Interleukin, TNFA: Tumor Necrosis Factor-α, IFNG: Interferon-γ, SEL: Socioeconomic Level.

**Table 1 ijms-24-13851-t001:** Baseline characteristics and clinical data of pregnant women included in the study.

	Low Risk for Preterm Delivery (*n* = 40)	High Risk for Preterm Delivery (*n* = 20)	*p*-Value
Age (years)	29 (±7.1)	31 (±5.8)	0.25
Pregestational weight (Kg)	63.7 (±13.7)	67.8 (±13.5)	0.08
Pregestational BMI (Kg/m^2^)	25.2 (±5.4)	27.5 (±5.3)	0.12
Socio-economic level, Median (Minimum and maximum value)	2 (1–4)	2 (1–5)	0.12
Smoking *n* (%)	1 (2.5)	0 (0)	0.45
History of preterm delivery *n* (%)	0 (0)	8 (40)	0.01 **
Gestational age at time of cervical length measurement, (weeks of gestation)	21.0 (±1.5)	21.2 (±2.0)	0.25
Cervical length (mm)	33.8 (±5.8)	13.1 (±7.7)	0.02 *
SPB < 28 WG *n* (%)	0	2 (10%)	0.001 ***
SPB 28–34 WG *n* (%)	2 (5%)	6 (30%)	0.001 ***
SPB > 34 WG *n* (%)	1 (2.5%)	1 (5%)	0.01 **

WG: weeks of gestation, SPB: spontaneous preterm birth. Mean, median, standard deviation, minimum and maximum value, comparisons with Mann–Whitney U and Chi-square Test, and *p*-value. * *p* < 0.05, ** *p* < 0.01, *** *p* < 0.001.

**Table 2 ijms-24-13851-t002:** Pro-inflammatory and anti-inflammatory cytokine profile in cervical-vaginal fluid at 18.0–23.6 weeks of gestation in high- (*n* = 20) and low-risk (*n* = 40) groups for preterm birth.

Cytokine	Risk Group for Preterm Birth	Mean ± SD pg/mL	*p*-Value
**Pro-inflammatory cytokines**			
**IL-1β**	High Risk	763.87 (±1505.99)	0.814
	Low Risk	587.94 (±1432.56)	
**IL-2**	High Risk	5.63 (±1.48)	0.01 **
	Low Risk	3.60 (±6.07)	
**IL-6**	High Risk	856.29 (±1.98)	0.001 ***
	Low Risk	118.32 (±0.48)	
**IL-8**	High Risk	5882.35 (±5638.79)	0.381
	Low Risk	9695.78 (±11,070.29)	
**IL-12**	High Risk	0.49 (±0.49)	0.304
	Low Risk	0.34 (0.29)	
**TNF-α**	High Risk	104.17 (±74.62)	0.115
	Low Risk	78.63 (±50.32)	
**IFN-γ**	High Risk	117.49 (±53.42)	0.001 ***
	Low Risk	54.17 (±26.37)	
**Anti-inflammatory cytokines**			
**IL-4**	High Risk	20.98 (±10.78)	0.001 ***
	Low Risk	10.83 (±8.92)	
**IL-10**	High Risk	40.44 (±41.23)	0.001 ***
	Low Risk	3.56 (±5.22)	
**IL-1ra**	High Risk	29,768 (±17,596)	0.002 ***
	Low Risk	58,377 (±40,841)	

Mean, standard deviation, comparisons with Mann–Whitney U Test, and *p*-value. ** *p* < 0.01, *** *p* < 0.001.

**Table 3 ijms-24-13851-t003:** Comparison among the two proposed classifiers models: “full model”, “adjusted model”, and the Fetal Medicine Foundation (FMF) calculator.

Random Forest “Full Model”			Random Forest “Adjusted Model”			Fetal Medicine Foundation Calculator		
Predicted	Real		Predicted	Real		Predicted	Real	
	Term	Preterm		Term	Preterm		Term	Preterm
Term	14	6	Term	20	1	Term	36	4
Preterm	2	1	Preterm	2	7	Preterm	7	13
Detection rate	65%		Detection rate	87.7%		Detection rate	79%	
False positive rate	12%		False positives rate	3.33%		False positives rate	6.6%	
False negative rate	28%		False negatives rate	6.66%		False negatives rate	11.66%	

## Data Availability

The data presented in this study are available on request from the corresponding author.

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
