# Peer review of "A Novel Predictive Machine Learning Model Integrating Cytokines in Cervical-Vaginal Mucus Increases the Prediction Rate for Preterm Birth"

_ijms, 2023, doi:10.3390/ijms241813851_

Round 1

Reviewer 1 Report

the manuscript is interesting and generally well written. Only minor points deserve to be improved. In particular:

Introduction: It deserves to be pointed out that Preterm Birth is associated to preterm premature rupture of membranes (PPROM). This is an important point to add since inflammatory cytokines play a key role in PPROM onset (as demonstrated here PMID: 26739007, PMID: 24768095)

Table 1 and 2: Statistical significant differences should be written in bold

Figure 1: Measure units on y axis must be added

4.4. Cervical-vaginal cytokine quantification: Product code must be added

Acronyms must be written in full length when mentioned for the first time

Author Response

Thank you for your comments on our manuscript "A Novel Predictive Machine Learning Model Integrating Cytokines in Cervical-Vaginal Mucus Increases the Prediction Rate for Preterm Birth" (ijms-2599982). We have addressed all your comments, which contributed to making our manuscript significantly better.

Reviewer 1

Reviewer: Introduction: It deserves to be pointed out that Preterm Birth is associated to preterm premature rupture of membranes (PPROM). This is an important point to add since inflammatory cytokines play a key role in PPROM onset (as demonstrated here PMID: 26739007, PMID: 24768095.)

Answer: the information and references requested were added in the introduction section, emphasizing the pathophysiological relationship between preterm premature rupture of membranes, preterm labor, and cytokine production:

Lanes 67-74: There is a pathophysiological association between preterm premature rupture of membranes (PPROM) and PB, both of which are triggered when the intrauterine or maternal environment is hostile. Moreover, PPROM is among the leading causes of preterm birth [14], probably due to the onset of an intrauterine infectious/inflammatory process, which produces an imbalance in cytokine production, disrupting the tight junctions of the membranes and rupturing the amniotic sac [15]. Some cytokines associated with PPROM and PB in placental tissue are IL-1β, IL-6, IL-8, and transforming growth factor (TGF-β) [16].

Reviewer: Table 1 and 2: Statistically significant differences should be written in bold

Answer: statistically significant differences in Tables 1 and 2 are now in bold.

Reviewer: Figure 1: Measure units on y axis must be added

Answer: The units on the y-axis (pg/mL) were added in Figure 1.

Reviewer: 4.4. Cervical-vaginal cytokine quantification: Product code must be added

Answer: The catalog number has been added to the kit used: Bio-Plex Pro Human Cytokine 10-plex (Cat 12020756), line 299

Reviewer: Acronyms must be written in full length when mentioned for the first time.

Answer: The complete legend was added to the acronyms when they were first used.

Dr. Hector Borboa-Olivares / Dr. Guadalupe Estrada-Gutierrez

Corresponding authors

Reviewer 2 Report

This observational cohort study investigated the screening potential of the cytokine profile in cervical-vaginal mucus in pregnant women with high- and low-risk for preterm birth based on obstetric history and cervical length. Furthermore, they then integrated their data in a screening model, using a predictive machine learning analysis.  The manuscript is well structured, and a pleasure to read. Overall, the topic is very relevant, and deserves to be publishedHowever, the manuscript suffers from unclear endpoints and ill-defined comparisons. Specific comments that have to be addressed: 
  1. 1. General:
  2. 1.1 There are some syntax glitches, and the manuscript could benefit from some copyediting 
  1. 2. Introduction:
  2. 2.1 Line 43-46. This first sentence of the introduction is very difficult to read. Please rewrite with a clear message and break up into clear concepts.
  3. 2.2 Line 73. The authors compare their work to the FMF calculator and state that approximately 70% preterm birth can be detected. The authors don’t define spontaneous preterm birth and this estimate is inaccurate, please use a reference for a specified outcome i.e., extreme (< 28 weeks), early (28–30 weeks), moderate (31–33 weeks) and mild (34–36 weeks) preterm birth.
  4. 2.3 Line 82. The authors quote that preventative strategies have an efficacy of up to 90%, citing a review article. This is not true and needs to be corrected i.e., in a program of cervical length screening and progesterone use there is up to a 44% reduction in preterm birth before 34w. 
  1. 3. Results
  2. 3.1 Line 92. Please define preterm birth. What gestational age was used and was this spontaneous preterm birth or iatrogenic?
  3. 3.2 Table 1. Include GA at delivery with breakdown of who delivered spontaneously before 28w, 34w and 37w.  
  4. 3.3 Table 2. There is no value for IL-1ra low risk.
  5. 3.4 Figure 1. Please define the y-axis value. Please define the plots, state the number of data points for each plot. These plots don’t correlate with table 2.
  6. 3.5 Figure 2. What is the need for the duplication of the predictors analyzed i.e., B is already incorporated in A.  
  1. 4. Methodology: 
  2. 4.1 Selection: it is unclear on how the patients were selected. On what basis was patients identified and selected.
  3. 4.2 According to the local protocol, high risk patients already received micronized progesterone but at what point was this initiated. Was these patietns already receiving progesterone at study entry?
  4. 4.3 Statistical analysis: was there any analysis done for data distribution? On what basis was the group size determined? Was there any power calculation done?  
  5. 4.4 Nowhere in the methods is preterm birth defined. What gestational age cutoffs were used and were these cases only spontaneous preterm births or were there also iatrogenic preterm birth included?

See above 

Author Response

Thank you for your comments on our manuscript "A Novel Predictive Machine Learning Model Integrating Cytokines in Cervical-Vaginal Mucus Increases the Prediction Rate for Preterm Birth" (ijms-2599982). We have addressed all your comments, which contributed to making our manuscript significantly better.

Reviewer 2

Reviewer:

  1. General:

1.1 There are some syntax glitches, and the manuscript could benefit from some copyediting.  

Answer: A revision of the manuscript was carried out in order to review the syntax glitches that were found.

Reviewer:

  1. Introduction:

2.1 Line 43-46. This first sentence of the introduction is very difficult to read. Please rewrite with a clear message and break up into clear concepts.

Answer: the wording of the sentence was corrected to read as follows: Preterm birth is a major public health issue concerning perinatal mortality, long-term morbidity, and economic burden, with a worldwide prevalence of 5-18%, 90% of which occurs in developing countries [1,2]

Reviewer:

2.2 Line 73. The authors compare their work to the FMF calculator and state that approximately 70% preterm birth can be detected. The authors don’t define spontaneous preterm birth and this estimate is inaccurate, please use a reference for a specified outcome i.e., extreme (< 28 weeks), early (28–30 weeks), moderate (31–33 weeks) and mild (34–36 weeks) preterm birth.

Answer: Definition of spontaneous preterm birth was added as follows:  Spontaneous preterm birth (SPB) is defined as the delivery of the fetus before 37 weeks’ gestation, calculated by the last menstrual period or reliable first-trimester ultrasound, without any medical intervention to induce this outcome [3] (Lanes 45-48).

The detection rate of the FMF calculator was change as follows: The calculator that is considered the gold standard in clinical practice is the Fetal Medicine Foundation (FMF) calculator, having a detection rate for preterm spontaneous birth <28 weeks = 75%, 28-30 weeks = 57%, 31-33 weeks = 46%, and 34-36 weeks = 24%, considering a false positive rate of 10% [17] (Lanes 81-82)

Reviewer:

2.3 Line 82. The authors quote that preventative strategies have an efficacy of up to 90%, citing a review article. This is not true and needs to be corrected i.e., in a program of cervical length screening and progesterone use there is up to a 44% reduction in preterm birth before 34w. 

Answer: the percentage of prevention with the use of progesterone was corrected to read as follows:

The goal of identifying women at high-risk for developing PB is to individualize the clinical follow-up and offering medical preventive strategies (e.g., progesterone) that reduces by up to 90% the risk of preterm birth in women with a history of this outcome and by 42% in pregnant women with short cervix detected in second trimester screening [20]. (Lanes 91-93)

Reviewer:

  1. Results

3.1 Line 92. Please define preterm birth. What gestational age was used and was this spontaneous preterm birth or iatrogenic?

Answer: It was pointed out that we are talking about spontaneous preterm labor before 37 gestational weeks as follows: The low-risk group had a prevalence of spontaneous preterm birth before 37 gestation weeks of 7.5% versus 45% in the high-risk group. (Lanes 102-104)

Reviewer:

3.2 Table 1. Include GA at delivery with breakdown of who delivered spontaneously before 28w, 34w and 37w.

Answer: stratification of preterm delivery was done in Table 1 as requested.

Reviewer:

  • Table 2. There is no value for IL-1ralow risk.

Answer: the value for IL-1ra in low-risk population was added in Table 2.

Reviewer:

3.4 Figure 1. Please define the y-axis value. Please define the plots, state the number of data points for each plot. 

Answer: The y-axis (pg/mL), plots, and number of experiments were added on Figure 1.

Reviewer

Figure 2. What is the need for the duplication of the predictors analyzed i.e., B is already incorporated in A.  

Answer: From our perspective, the predictor variables are not duplicated. In section A of the figure, we show the “full model” with all the predictor variables studied. In section B, we show only the three variables with the highest statistical significance, the "adjusted model," to create a model with fewer variables that shows a good performance in terms of the detection rate. The second model is more reproducible and less costly, with a smaller number of variables. Therefore, we kindly ask the reviewer to consider keeping both models, as the performance of both was evaluated.

Reviewer

  1. Methodology: 

4.1 Selection: it is unclear on how the patients were selected. On what basis was patients identified and selected.

Answer: Sampling at convenience was performed until the total number of patients included in the study was reached. We added this information in Line 364 in the methodology section. Patients were selected according to the cervical length evaluated and classified as high or low risk for preterm labor based on this measure as described in the original manuscript (Lanes 270-272).

Reviewer

4.2 According to the local protocol, high risk patients already received micronized progesterone but at what point was this initiated. Was these patietns already receiving progesterone at study entry?

Answer: Treatment was initiated after being classified as a high-risk patient based on CL measurement. No patient was receiving progesterone before being evaluated by CL measurement. This information in now included in the manuscript (Lanes 283-287)

Reviewer

4.3 Statistical analysis: was there any analysis done for data distribution? On what basis was the group size determined? Was there any power calculation done?  

This table shows the statistical power for the cytokines with the highest significance (FYI only)

Answer: The following statements were included in the manuscript: Data distribution was verified before the statistical analysis with Kolmogorov–Smirnov test. Cytokine concentration was evaluated, and differences between groups were assessed using the Mann-Whitney U test. The Chi-square test was used to calculate the difference in proportions. No sample size calculation was performed beforehand, but the statistical power was calculated for all variables with significant differences to verify that it was greater than 80% (Lanes 306-312).

This table shows the statistical power for the cytokines with the highest significance (FYI only)

Reviewer

4.4 Nowhere in the methods is preterm birth defined. What gestational age cutoffs were used and were these cases only spontaneous preterm births or were there also iatrogenic preterm birth included?

Answer: we add in methods the definition of spontaneous preterm labor as primary outcome.  Spontaneous preterm birth is defined as the delivery of the fetus before 37 weeks’ gestation, calculated by the last menstrual period or reliable first-trimester ultrasound, without any medical intervention that produced this outcome. Lanes 277-280).

Dr. Hector Borboa-Olivares / Dr. Guadalupe Estrada-Gutierrez

Corresponding authors
